# Sex-Specific Energy Intakes and Physical Activity Levels According to the Presence of Metabolic Syndrome in Korean Elderly People: Korean National Health and Nutrition Examination Survey 2016–2018

**DOI:** 10.3390/ijerph17155416

**Published:** 2020-07-28

**Authors:** Won-Sang Jung, Hun-Young Park, Sung-Woo Kim, Kiwon Lim

**Affiliations:** 1Physical Activity and Performance Institute (PAPI), Konkuk University, 120 Neungdong-ro, Gwangjin-gu, Seoul 05029, Korea; jws1197@konkuk.ac.kr (W.-S.J.); parkhy1980@konkuk.ac.kr (H.-Y.P.); kswrha@konkuk.ac.kr (S.-W.K.); 2Department of Sports Medicine and Science, Graduate School, Konkuk University, 120 Neungdong-ro, Gwangjin-gu, Seoul 05029, Korea; 3Department of Physical Education, Konkuk University, 120 Neungdong-ro, Gwangjin-gu, Seoul 05029, Korea

**Keywords:** energy intake level, physical activity level, metabolic syndrome, elderly people

## Abstract

This study aimed to analyze the differences in energy intake and physical activity (PA) levels according to sex and the presence of metabolic syndrome (MetS) among elderly people in Korea. Data of 3720 elderly people (aged >65 years) were obtained from the Korean National Health and Nutrition Examination Survey (2016–2018). We analyzed PA levels (occupational and recreational PA) and energy intakes (carbohydrate, protein, and fat). The MetS group showed lower levels of moderate intensity recreational PA and place movement than the non-MetS group (*p* < 0.05); in the MetS group, PA levels were significantly lower in women than in men (*p* < 0.05). The intakes of total energy, carbohydrate, fat, and protein were lower in the MetS group than in the non-MetS group (*p* < 0.001). Both the non-MetS and MetS groups showed lower energy intakes in women than men (*p* < 0.001). Our study shows that elderly people, especially women, with MetS have significantly lower total PA levels and total energy intakes. We confirmed the importance of increased PA and proper nutritional intake in elderly people. Therefore, it is believed that practical measures such as nutrition education and nutrition guidance and PA education are urgently needed to reduce the incidence of MetS among the elderly.

## 1. Introduction

In modern societies, the increase in average life expectancy due to advances in medical technology and improvements in economic status has resulted in an increasingly aging population. Increased body fat and lower muscle mass as a result of aging cause physical problems in this population [1,2,3]. Furthermore, considering the deterioration in quality of life and increase in socioeconomic problems, such as chronic morbidity, medical expenses, and mortality, it is important to find ways to solve the health problems of elderly people [2,4].

The incidence of obesity, diabetes, high blood pressure, and cardiovascular diseases among elderly people is high, and metabolic syndrome (MetS), a major risk factor for chronic diseases, is on the rise [5]. The Korean Society of Cardio Metabolic Syndrome reported that senior citizens aged >65 years had a MetS rate of about 35% [6,7]. The prevalence of MetS (obesity, high blood pressure, and elevated high-density lipoprotein cholesterol (HDL-C) levels), which is a risk factor for cardiovascular diseases, has been reported to be significantly higher among elderly people in Korea than in those in the United States and Europe [8,9].

The increased risk of MetS in elderly people is mainly due to changes in body composition: an increase in body fat mass and a decrease in free fat mass and muscle strength due to aging [10]. Elderly people with high-calorie diets and decreased physical activity (PA) levels show a gradual increase in body fat due to excessive accumulation of body fat, and metabolic factors such as obesity, diabetes, and inflammation increase the risk of cardiovascular diseases [6,11,12]. In particular, a decrease in muscle mass and an increase in body fat were found to have a negative effect on the condition of the metabolism, such as decreased body function, physical strength, blood lipids, body inflammation, and hormones [2,13,14].

MetS is reported to be highly correlated with type 2 diabetes, cerebrovascular diseases, cardiovascular diseases, and cancer (colorectal, pancreatic, and breast cancer), which are the main causes of mortality among elderly people in South Korea [15,16,17]. These typical risk factors are associated with high levels of C-reactive protein and homocysteine, which indicate a pro-inflammatory state in the blood vessels [18]. The pro-inflammatory state causes hyper-viscosity by activating blood coagulation reactions, increasing the level of plasminogen activator inhibitor (PAI)-1 or fibrinogen [19], and increasing the level of low-density lipoprotein cholesterol (LDL-C). Additionally, the resting heart rate increases due to stress, activating the sympathetic nervous system and decreasing the activation of the parasympathetic nervous system [20,21]. Experimental studies have shown that these risk factors (obesity, type 2 diabetes, hypertension, etc.) for the prevalence of MetS can be modulated by increased PA and continuous exercise [22,23]. However, there is a paucity of research on the efficiency of PA that is based on a large sample of elderly people of both sexes.

Risk factors for MetS have been reported to also be closely related to diet and nutrition, PA levels, and exercise participation [11]. Nutrition in elderly people is influenced by a variety of factors, including socioeconomic levels, physical health, and emotional state and also acts as a major determinant of health [8]. Elderly people are prone to poor nutrition due to poor physiological function, poor digestion, poor dental conditions, economic difficulties after retirement, and depression in old age. Poor nutrition is associated with increased health risks such as decreased immunity, decreased physical function, and an increased morbidity rate [24,25]. In particular, fat and carbohydrate intakes are closely related to MetS, while the intake of essential fatty acids, calcium, magnesium, and dairy products has been reported to reduce the risk of and lower the prevalence rate of MetS [26,27]. Malnutrition in elderly people has been associated with lower income and education levels and with activity restriction or depression. Eating balanced meals can prevent the occurrence of cardiovascular disease, diabetes, hypertension, etc. and chronic degenerative diseases such as osteoporosis [28,29].

The level of PA, the degree of participation in exercise, and dietary and nutritional intakes have an impact on the risk factors and prevalence of MetS. Considering that these factors vary according to an elderly person’s lifestyle and sex, it is very important to examine the relationship between these factors in a large population in Korea. Moreover, cohort studies with an adequate sample size evaluating PA levels and energy intakes according to sex and the presence of MetS are insufficient. Therefore, this study aimed to analyze the differences in energy intake and PA levels by sex and the presence of MetS in a Korean elderly population, based on data from the 7th Korea National Health and Nutrition Examination Survey (2016–2018) and to investigate the relationship.

## 2. Materials and Methods

### 2.1. Sample and Design

This study used cross-sectional data from the Korea National Health and Nutrition Examination Survey (KNHANES) from 2016 to 2018, which was conducted by the Korea Centers for Disease Control and Prevention (KCDC). The details of the study design and data resource profiles followed the methods in the Guidelines for Use of the KNHANES Raw Data and the Final Report of sampling frame [16]. The KNHANES consists of a health interview survey, a nutrition survey, and a health examination and is conducted according to the Declaration of Helsinki. This survey was approved by the Institutional Review Board of the Korea Centers for Disease Control and Prevention (reference number: 2018-01-03-P-A). All participants in the survey signed an informed consent form.

Between 2016 and 2018, 20,659 individuals completed the health interview survey, nutrition survey, and health examination. Among them, 15,703 people under 65 years of age were excluded, leaving 4965 people over 65 years of age. Participants previously diagnosed with and/or treated for cancer (gastric, liver, colon, breast, cervical, lung, thyroid, and other cancers), those who had undergone surgery for other indications, and those with missing data (anthropometric, health examination, and PA data) were excluded (Figure 1). In total, 3720 elderly people were finally included in this study.

### 2.2. Measures

The analytical items of this study were the height and body weight of the screening survey items from the KNHANES Raw Data. The presence or absence of MetS was determined using measurements of waist circumference, blood pressure, fasting blood glucose levels, triglyceride levels, and HDL-C levels. PA variables were evaluated using the Global Physical Activity Questionnaire (GPAQ), and PA was expressed in metabolic equivalents (MET)-minutes/week. Nutrient intake and intake rates were also analyzed. The analyzed characteristics of the participants according to sex are shown in Table 1.

### 2.3. Metabolic Syndrome

The diagnosis of MetS was based on the new harmonized guidelines of the National Cholesterol Education Program-Adult Treatment Panel III and the American Heart Association and the National Heart Lung and Blood Institute [30,31]. For waist circumference, we followed the criteria suggested by the Korean Society for the Study of Obesity [32]. If three or more of the following five criteria were met, participants were classified as having MetS: waist circumference > 90 cm (men) or >85 cm (women); systolic blood pressure > 130 mmHg or diastolic blood pressure > 85 mmHg; fasting triglyceride (TG) levels > 150 mg/dL; fasting HDL-C levels < 40 mg/dL (men) or <50 mg/dL (women); and fasting glucose levels > 100 mg/dL.

### 2.4. Physical Activity

The GPAQ comprises 16 questions grouped to capture PA undertaken in different behavioral domains: work, transport, and discretionary (also known as leisure or recreational) activities. It analyzes five domains of PA: vigorous-intensity work, moderate-intensity work, place movement, vigorous-intensity recreation, and moderate-intensity recreation. The World Health Organization (WHO) GPAQ analysis guidelines were used to analyze GPAQ data [33]. We estimated that, compared to while sitting quietly, a person’s caloric consumption was four times higher when they were being moderately active and eight times higher when they were being vigorously active. Therefore, when calculating the total energy consumption of an individual using GPAQ data, four METs were allocated to the time spent in moderate activity and eight METs to the time spent in vigorous activity, and the details are as follows:Vigorous intensity activity: occupational (MET) = 8.0 × vigorous intensity physical activity (day/week) × 1-day vigorous intensity physical activity (minutes/day)Moderate intensity activity: occupational (MET) = 4.0 × moderate intensity physical activity (day/week) × 1-day moderate intensity physical activity (minutes/day)Vigorous intensity activity: recreational (MET) = 8.0 × vigorous intensity physical activity (day/week) × 1-day vigorous intensity physical activity (minutes/day)Moderate intensity activity: recreational (MET) = 4.0 × moderate intensity physical activity (day/week) × 1-day moderate intensity physical activity (minutes/day)Place movement (MET) = 4.0 × place movement physical activity (day/week) × 1-day place movement physical activityTotal Physical Activity (MET) = vigorous intensity activity: occupational + moderate intensity activity: occupational + vigorous intensity activity: recreational + moderate intensity activity: recreational + place movement.

PA levels were classified into four groups: inactive (0–249 MET min/week), somewhat active (250–499 MET min/week), active (500–999 MET min/week), and very active (>1000 MET min/week). These cut-off points are based on their equivalence to the following PA thresholds: 250 MET min/week represents an energy expenditure dose equivalent to half of threshold; 500 MET min/week is equivalent to the minimal threshold; and 1000 MET min/week is equivalent to twice the minimal threshold.

### 2.5. Energy Intake and Intake Ratio

The nutrition survey of the KNHANES consisted of a survey of dietary habits, 1-day 24-hour recall, and food frequency questionnaire administration. The nutrition survey data were collected by trained dietitians in the homes of the participants 1 week after the health interview and health examination. The daily energy intake was calculated using the Korean Foods and Nutrients Database of the Rural Development Administration. The following items were included in the analyses: calorie intake, macronutrient (carbohydrates, proteins, fat) intake, and energy ratio.

### 2.6. Statistical Analysis

Continuous variables were presented as means and standard errors. Normality of distribution of all outcome variables were verified using the Kolmogorov–Smirnov test. A two-way analysis of variance (ANOVA) was used to analyze the differences in PA levels, energy intakes, intake rates, risk factors for MetS between participants with and without MetS, and between men and women. Partial eta-squared (*η*^2^) values were calculated as measures of the effect size. If a significant interaction effect was found, post-hoc independent t-tests were used to compare the presence and absence of MetS and the sex-specificity of dependent variables in each group separately. Moreover, the relationships between PA levels and MetS were determined using logistic regression after controlling for covariates; logistic regression findings were presented as odds ratios (ORs) and their associated 95% confidence intervals (CIs). Statistical analyses were performed using IBM Statistical Package for Social Science (SPSS) version 25.0 for Windows (IBM Corp., Armonk, NY, USA); *p* < 0.05 was considered statistically significant.

## 3. Results

### 3.1. Differences in Physical Activity Levels According to the Presence or Absence of Metabolic Syndrome and According to Sex

The differences in PA levels according to the presence or absence of MetS and according to sex are presented in Table 2. Elderly individuals without MetS had significantly higher PA levels than did those with MetS (*p* < 0.001; moderate intensity activity: recreational (*p* = 0.002) and total PA (*p* = 0.006)). When analyzed separately, men did not show significant differences in PA levels based on the presence or absence of MetS (*p* > 0.05); however, in women, individuals without MetS had significantly higher levels of total PA, (*p* = 0.010), place movement (*p* = 0.002), and moderate intensity activity: recreational (*p* = 0.001) than those with MetS.

Women had significantly lower levels of place movement (*F* = 6.889, *p* = 0.009), vigorous intensity activity: recreational (*F* = 22.790, *p* < 0.001), moderate intensity activity: recreational (*F* = 59.790, *p* < 0.001), and total PA (*F* = 34.916, *p* < 0.001) than did men, regardless of the presence or absence of MetS.

### 3.2. Difference in Energy Intakes According to the Presence or Absence of Metabolic Syndrome and According to Sex

Table 3 shows the differences in energy intake according to the presence or absence of MetS and according to sex. Elderly participants without MetS had significantly higher intakes of total energy (1677 vs. 1572 kcal; *p* < 0.001), carbohydrates (*p* < 0.001), proteins (*p* < 0.001), and fats (*p* < 0.001) than those with MetS. Among women, participants without MetS had significantly higher intakes of total energy (*p* = 0.001), carbohydrates (*p* = 0.008), proteins (*p* < 0.001), and fats (*p* = 0.001), while in men, there was no significant difference in energy intake regardless of the presence or absence of MetS. Looking at the differences in sex, female participants had significantly less energy intake in all variable (*p* < 0.001), irrespective of whether they had MetS or not.

### 3.3. Odds Ratios (95% CI) for MetS and MetS Components According to Physical Activity Levels

The average values according to PA levels are shown in Table 4 and the odds ratio values for the MetS and MetS components according to PA levels are presented in Table 5. We found that the higher the PA level, the lower were the values for waist circumference, TG, blood pressure, and blood glucose and the higher were the values for HDL-C. When the PA activity level was above the active levels, waist circumference values significantly decreased compared to the reference value (OR = 0.80, 95% CI = 0.66–0.96), while TG (OR = 0.79, 95% CI = 0.65–0.97) and blood pressure (OR = 0.82, 95% CI = 0.69–0.99) were significantly decreased in very active levels, and HDL-C (OR = 0.64, 95% CI = 0.53–0.77) increased. Blood glucose levels decreased with increasing PA levels, but there was no significant difference.

## 4. Discussion

In this study, we analyzed the differences in sex-specific dietary intakes and PA levels according to the presence or absence of MetS in elderly people based on data from the KNHANES. Elderly people with MetS had low PA levels and insufficient energy intakes. Elderly women had significantly lower PA levels and energy intakes and had nutritional imbalance, demonstrated by the consumption of high amounts of carbohydrates and low amounts of fat. To prevent MetS and MetS components related variables, PA levels above the active (500–999 MET min/week) or very active (>1000 MET min/week) level are required. These results suggest that to prevent MetS in elderly people, PA levels should be increased, and balanced nutrition should be assured. In particular, PA programs and education regarding balanced nutrition for elderly women may be necessary for long-term health promotion.

According to the KNHANES, the incidence of MetS in Koreans has increased by 0.6% every 10 years since 1998 [34]. This increase reflected the rapid recovery of the Korean society after the economic crisis and also reflected the rapid change in lifestyle, especially the adoption of high fat and high carbohydrate intake and low physical activity, which has contributed to the increase in MetS incidence. Total energy intake, especially fat and carbohydrate intakes, has been steadily increasing since 2008. Conversely, the amount of PA has decreased, with a pattern toward a decrease in moderate PA and walking rather than an increase in high-intensity PA [35].

In particular, elderly people with sarcopenia, increased body fat, and decreased muscle strength and muscle endurance due to aging have been found to have a high incidence of MetS due to reduced activities of daily living or the presence of chronic diseases [2,36], which in turn leads to a decrease in quality of life due to limited PA such as reduced PA as a result of cardiovascular diseases and reduced muscular strength, muscular endurance, and balance [37,38].

According to a recent report on the National Health and Nutrition Examination Survey, there is a significant sex-difference in MetS incidence. In men, the incidence of MetS increased with age from the 20s to 50s, but it did not increase thereafter. In contrast, the incidence rate in women increased until the 7th decade, especially in the 60s and 70s. Additionally, an analysis of MetS components showed that high blood pressure in women aged >50 years increased rapidly, and that the increase in blood pressure and blood glucose levels in men were closely related to the increase in total energy intake, carbohydrates, and fat intake [39,40]. Therefore, in this study, the relationships between MetS and PA levels and dietary intakes in elderly people were stratified by sex.

Overall, the amount of PA according to the presence or absence of MetS and according to sex was different in terms of recreational PA and place movement rather than occupational PA. Amagasa et al. [41] reported that the levels of moderate and vigorous intensity PA in elderly people in Japan did not differ significantly according to sex and that men who weighed more consumed more calories through PA than women. In the study by Davis and Fox [42], men aged >70 years had significantly higher PA levels than women, which is consistent with our findings. Participation in moderate and vigorous intensity PA is known to have a positive effect on the risk factors and incidence of MetS [36]. A study by Laaksonen et al. [43] reported that vigorous intensity PA (>60 min per week) was associated with a lower incidence of MetS (63% lower) after adjusting for age and body mass index. As such, we found that to reduce the incidence of MetS, PA equivalent to more than active PA (500–999 MET min/week) must be performed. Zbronska et al. [44] reported that 1500 MET min/week PA could bring benefits to health-related variables. This study also showed that highly active (>1000 MET min/week) PA is needed to improve the variables related to MetS. Schaller et al. reported that not only the intensity of PA but also the type of PA could affect the quality of health-related life. Pedišić et al. [45] reported that the quality of life decreased as occupational PA levels increased in men and women. In all, it was confirmed that health-related quality of life improved as the levels of recreational PA and place movement increased.

An analysis of dietary intake according to the presence or absence of MetS in elderly people showed that total energy intake and carbohydrate, fat, and protein intakes were significantly higher in the non-MetS group, with similar patterns being witnessed on stratification by sex. However, significant differences were found in elderly women. Similar studies have shown that elderly adults have poor nutrition, with higher levels of carbohydrate intake and lower levels of protein and fat intakes than younger age groups [46]. Studies on elderly people in our country have shown that they have nutritional levels below the nutritional recommendations and that the nutritional status of elderly people worsens with age [47,48,49]. In particular, malnutrition in elderly adults is known to cause a decrease in immune function and inflammatory reactions, increasing the risk of chronic diseases [48]. In this study, we suggest that lower intake may have facilitated more inflammatory reactions for reasons of lower calorie intake [50]. In this study, elderly people with MetS weighed more than elderly people without MetS, but given the low calorie intake, it is possible that the inflammatory reaction may have been facilitated.

According to the Korean nutritional standards, the recommended energy sources for people aged > 19 years should be as follows: 55–70% carbohydrates, 7–20% protein, and 15–25% fat. In this study, the proportion of carbohydrate intake was high and the proportions of protein and fat were low in both the MetS and non-MetS groups; these differences were bigger in the MetS group. This trend was similar between older men and older women, but significantly stronger in women.

In a prior study, it was reported that rice and mixed grains are the main energy sources of Koreans, and that there is a problem with the variety of side dishes. It was also reported that elderly people with diabetes consume a lot of carbohydrates and avoid foods with high fat content, which is why the nutritional imbalance is higher [51]. In Europe, on the other hand, high energy consumption and increased intake of simple carbohydrates and fat are reported, which seems to be different from characteristics of elderly Korean people caused by nutritional imbalance [52]. As a result, the health and nutrition problems of the elderly Korean are increasing due to the lack of energy intake and poor micronutrient intake. A prior study said that the choice of diet was not due to insufficient resources, but due to lack of information and food intake habits, and that food intake habits were established later in life [53]. These findings suggest the importance of developing nutrition management programs to optimize the dietary intakes of elderly people with MetS, especially women, so as to maintain an optimal nutritional status.

MetS is caused by the effects of lifestyle and genetic factors such as diet, weight, and physical activity, and it is reported that the risk of MetS can be reduced due to balanced eating habits and increased PA [54,55,56]. In elderly participants with MetS, malnutrition rather than an excess of nutrition appears to be the problem. Consumption of a balanced diet is more important than the lack of intake of a single nutrient intake. Therefore, future studies should evaluate and recommend best eating practices that can reduce the risk of MetS among elderly Korean people. Elderly people with MetS are also thought to have higher weight, consume less nutrients (including calories), and have lower PA levels; thus, it is believed that this information can be used as basic data in preparing countermeasures through nutrition education, nutrition guidance, and PA education for the elderly with MetS.

Combining the results of this and prior studies, the nutritional status of the elderly with MetS is poor, and the opportunity to eat quality meals and perform PA is low. Therefore, it is believed that practical measures such as nutrition education, nutrition guidance, and PA education are urgently needed to reduce the incidence of MetS among the elderly.

Based on the unique correlation between MetS and PA levels and nutrition in elderly people, which has been previously reported, the differences noted in occupational PA, recreational PA, and nutrition according to the presence or absence of MetS and according to sex in this study can be used as a basis for developing lifestyle modifications that can be applied uniquely to Korean elderly people. In particular, education and management regarding balanced nutrition are needed to address the increased carbohydrate intake and decreased fat and protein intakes seen among elderly people. Additionally, it is believed that institutional and policy support will help elderly people with MetS to safely engage in intensive PA.

This study should be interpreted considering the following limitations. First, we evaluated elderly people with MetS but did not consider the timing of MetS development or the duration of MetS. Second, the amount of PA was not assessed using heart rate measurements or using an accelerometer but quantified based on survey findings, which are prone to errors. Finally, this study reported simple differences without identifying the causality underlying the relationship between PA and nutrition.

## 5. Conclusions

Our study shows that elderly individuals with MetS have significantly reduced PA levels and have decreased levels of total energy and macronutrient intakes. In particular, elderly women with MetS have low PA levels and energy intakes. These findings highlight the importance of increased PA and proper nutritional intake in elderly people, especially in women. Programs aimed at increasing PA levels and education and care regarding balanced nutrition are needed.

## Figures and Tables

**Figure 1 ijerph-17-05416-f001:**
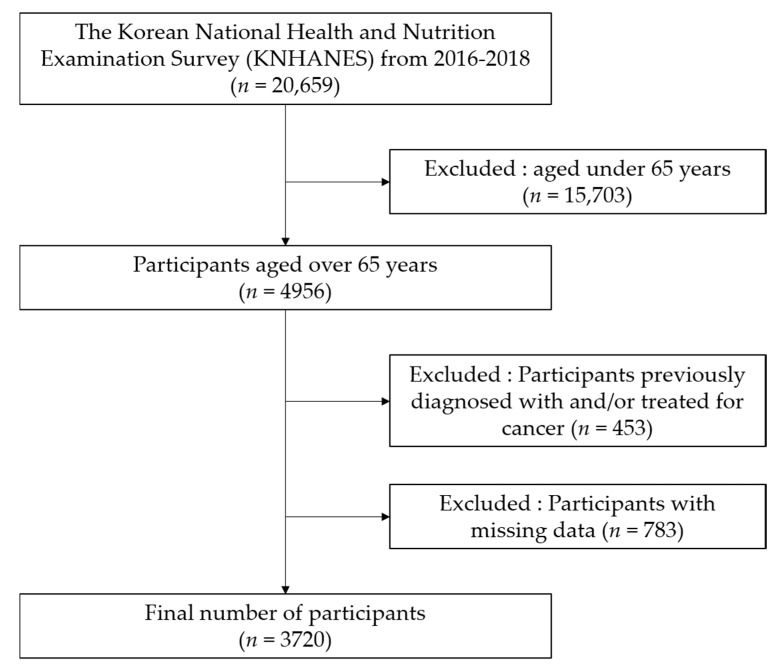
Flow diagram for the selection of study participants.

**Table 1 ijerph-17-05416-t001:** Descriptive characteristics of the participants.

Variables	Total (*n* = 3720)		Male (*n* = 1586)		Female (*n* = 2134)	
	Non-MetS (*n* = 2347)	MetS (*n* = 1373)	Non-MetS (*n* = 1018)	MetS (*n* = 586)	Non-MetS (*n* = 1248)	MetS (*n* = 886)
Age (years)	72.8 ± 0.1	73.0 ± 0.1	72.7 ± 0.2	72.4 ± 0.2	72.8 ± 0.2	73.3 ± 0.2
Height (cm)	158.2 ± 0.2	157.0 ± 0.2	165.3 ± 0.2	165.9 ± 0.2	151.7 ± 0.2	151.9 ± 0.2
Body weight (kg)	57.7 ± 0.2	63.4 ± 0.2	62.4 ± 0.3	69.8 ± 0.4	53.4 ± 0.2	59.7 ± 0.3
BMI (kg/m^2^)	23.0 ± 0.1	25.7 ± 0.1	22.8 ± 0.1	25.3 ± 0.1	23.2 ± 0.1	25.9 ± 0.1
Alcohol	75.3	70.5	90.5	90.6	61.7	59.4
Smoking	39.3	32.3	77.1	80.5	5.7	5.5
Waist circumference (cm)	81.7 ± 0.2	89.7 ± 0.2	83.7 ± 0.3	92.0 ± 0.3	80.0 ± 0.2	88.4 ± 0.2
TG (mg/dL)	102.6 ± 1.0	173.2 ± 2.5	103.1 ± 1.6	183.3 ± 3.8	102.1 ± 1.2	167.3 ± 3.2
HDL-C (mg/dL)	52.1 ± 0.3	42.8 ± 0.2	49.3 ± 0.4	40.4 ± 0.4	54.6 ± 0.4	44.1 ± 0.3
SBP (mmHg)	125.5 ± 0.4	132.4 ± 0.4	123.8 ± 0.5	130.2 ± 0.7	127.0 ± 0.5	133.6 ± 0.5
DBP (mmHg)	71.7 ± 0.2	73.0 ± 0.3	71.6 ± 0.3	72.4 ± 0.4	71.7 ± 0.3	73.4 ± 0.3
Fasting glucose (mg/dL)	100.3 ± 0.4	118.5 ± 0.8	102.6 ± 0.7	122.9 ± 1.5	98.2 ± 0.5	116.0 ± 0.9
Mets (%)	63.1	36.9	69.2	30.8	58.5	41.5
Occupational activity: vigorous	20.2 ± 7.6	35.6 ± 16.2	32.0 ± 13.9	71.1 ± 37.6	9.4 ± 7.0	15.2 ± 13.4
Occupational activity: moderate	55.0 ± 12.2	49.3 ± 10.6	55.3 ± 14.8	46.8 ± 14.4	54.62 ± 19.1	50.7 ± 14.6
Place movement	399.3 ± 15.3	322.4 ± 14.0	424.9 ± 24.4	362.9 ± 26.0	375.9 ± 18.9	299.2 ± 16.3
Recreational activity: vigorous	30.4 ± 7.2	26.2 ± 6.2	54.2 ± 14.5	57.0 ± 14.2	8.7 ± 3.4	8.5 ± 5.4
Recreational activity: moderate	108.2 ± 8.7	73.0 ± 7.1	148.0 ± 15.0	139.3 ± 16.9	71.7 ± 9.3	34.8 ± 5.3
Total physical activity	613.1 ± 25.3	506.4 ± 29.3	714.3 ± 40.5	677.1 ± 60.1	520.3 ± 30.9	408.3 ± 30.3
Total energy intake (kcal)	1677.3 ± 15.4	1572.1 ± 16.2	1898.2 ± 21.8	1891.9 ± 28.6	1475.1 ± 19.8	1388.3 ± 17.2
Carbohydrate intake (kcal)	1175.9 ± 10.9	1114.9 ± 10.9	1285.7 ± 14.8	1275.0 ± 18.3	1075.3 ± 15.3	1022.8 ± 12.8
Protein intake (kcal)	252.3 ± 4.3	227.7 ± 4.7	287.4 ± 6.4	288.7 ± 9.2	220.1 ± 5.5	192.6 ± 4.9
Fat intake (kcal)	225.7 ± 2.6	207.5 ± 2.7	260.3 ± 4.0	255.2 ± 5.0	194.0 ± 3.0	180.1 ± 2.8
Carbohydrate intake (%)	71.3 ± 0.3	72.6 ± 0.3	68.9 ± 0.4	69.2 ± 0.5	73.5 ± 0.3	74.5 ± 0.3
Protein intake (%)	13.4 ± 0.1	13.1 ± 0.1	13.6 ± 0.1	13.3 ± 0.1	13.1 ± 0.1	12.9 ± 0.1
Fat intake (%)	14.4 ± 0.2	13.7 ± 0.2	14.6 ± 0.2	14.4 ± 0.3	14.2 ± 0.2	13.2 ± 0.2

Values are expressed as means ± standard errors; BMI, body mass index; MetS, metabolic syndrome; TG, triglyceride; HDL-C, high-density lipoprotein cholesterol; SBP, systolic blood pressure; DBP, diastolic blood pressure; MET, metabolic equivalents.

**Table 2 ijerph-17-05416-t002:** Levels of physical activity.

Physical Activity (MET × min/Week)	Group	Total	Male	Female	*p*-Value	ANOVA			
						*F*-Value		*p*-Value (*η*^2^)	Power
Occupational vigorous	Non-MetS	20.2 ± 7.6	32.0 ± 13.9	9.4 ± 7.0	0.148	S	5.118	0.024 (0.001)	0.619
	MetS	35.6 ± 16.2	71.1 ± 37.6	15.2 ± 13.4	0.162	M	1.678	0.195 (0.000)	0.254
	*p*-value	0.364	0.329	0.704		S × M	0.926	0.336 (0.000)	0.161
Occupational moderate	Non-MetS	55.0 ± 12.2	55.3 ± 14.8	54.62 ± 19.06	0.977	S	0.009	0.925 (0.000)	0.051
	MetS	49.3 ± 10.6	46.8 ± 14.4	50.69 ± 14.56	0.861	M	0.136	0.713 (0.000)	0.066
	*p*-value	0.731	0.697	0.869		S × M	0.018	0.893 (0.000)	0.052
Place movement	Non-MetS	399.3 ± 15.3	424.9 ± 24.4	375.9 ± 18.9	0.112	S	6.889	0.009 (0.002)	0.747
	MetS	322.4 ± 14.0	362.9 ± 26.0	299.2 ± 16.3	0.038 *	M	10.439	0.001 (0.003)	0.898
	*p*-value	< 0.001 ***	0.082	0.002 **		S × M	0.116	0.733 (0.000)	0.063
Recreational vigorous	Non-MetS	30.4 ± 7.2	54.2 ± 14.5	8.7 ± 3.4	0.002 **	S	22.790	0.000 (0.006)	0.998
	MetS	26.2 ± 6.2	57.0 ± 14.2	8.5 ± 5.4	<0.001 ***	M	0.018	0.894 (0.000)	0.052
	*p*-value	0.661	0.894	0.974		S × M	0.024	0.877 (0.000)	0.053
Recreational moderate	Non-MetS	108.2 ± 8.7	148.0 ± 15.0	71.7 ± 9.3	<0.001 ***	S	59.790	0.000 (0.016)	10.000
	MetS	73.0 ± 7.1	139.3 ± 16.9	34.8 ± 5.3	<0.001 ***	M	3.811	0.051 (0.001)	0.497
	*p*-value	0.002 **	0.706	0.001 **		S × M	1.458	0.227 (0.000)	0.227
Total physical activity	Non-MetS	613.1 ± 25.3	714.3 ± 40.5	520.3 ± 30.9	<0.001 ***	S	34.916	0.000 (0.009)	10.000
	MetS	506.4 ± 29.3	677.1 ± 60.1	408.3 ± 30.3	<0.001 ***	M	3.634	0.057 (0.001)	0.478
	*p*-value	0.006 **	0.593	0.010 *		S × M	0.911	0.340 (0.000)	0.159

Values are expressed as means ± standard errors; MetS, metabolic syndrome. Main effect = S (sex) and M (metabolic syndrome); Interaction effect = S × M (sex × metabolic syndrome); * *p* < 0.05; ** *p* < 0.01; *** *p* < 0.001.

**Table 3 ijerph-17-05416-t003:** Energy intake.

Variables	Group	Total	Male	Female	*p*-Value	ANOVA			
						*F*-Value		*p*-Value (*η*^2^)	Power
Total energyintake (kcal)	Non-MetS	1677.3 ± 15.4	1898.2 ± 21.8	1475.1 ± 19.8	<0.001 ***	S	460.451	0.000 (0.110)	1.000
	MetS	1572.1 ± 16.2	1891.9 ± 28.6	1388.3 ± 17.2	<0.001 ***	M	4.652	0.031 (0.001)	0.578
	*p*-value	<0.001 ***	0.859	0.001 **		S × M	3.479	0.062 (0.001)	0.462
Carbohydrate intake (kcal)	Non-MetS	1175.9 ± 10.9	1285.7 ± 14.8	1075.3 ± 15.3	<0.001 ***	S	224.853	0.000 (0.057)	1.000
	MetS	1114.9 ± 10.9	1275.0 ± 18.3	1022.8 ± 12.8	<0.001 ***	M	4.188	0.041 (0.001)	0.534
	*p*-value	<0.001 ***	0.650	0.008 **		S × M	1.832	0.176 (0.000)	0.272
Fat intake (kcal)	Non-MetS	252.3 ± 4.3	287.4 ± 6.4	220.1 ± 5.5	<0.001 ***	S	165.840	0.000 (0.043)	1.000
	MetS	227.7 ± 4.7	288.7 ± 9.2	192.6 ± 4.9	<0.001 ***	M	4.255	0.039 (0.001)	0.541
	*p*-value	<0.001 ***	0.901	<0.001 ***		S × M	5.177	0.023 (0.001)	0.624
Protein intake (kcal)	Non-MetS	225.7 ± 2.6	260.3 ± 4.0	194.0 ± 3.0	<0.001 ***	S	376.450	0.000 (0.092)	1.000
	MetS	207.5 ± 2.7	255.2 ± 5.0	180.1 ± 2.8	<0.001 ***	M	6.834	0.009 (0.002)	0.743
	*p*-value	<0.001 ***	0.422	0.001 **		S × M	1.457	0.227 (0.000)	0.226

Values are expressed as means ± standard errors, MetS = metabolic syndrome. Main effect = S (sex) and M (metabolic syndrome); Interaction effect = S × M (sex × metabolic syndrome); ** *p* < 0.01, *** *p* < 0.001.

**Table 4 ijerph-17-05416-t004:** Classification of physical activity levels.

Physical Activity Level	MET min/Week (Mean ± SE)				
	Total	*n*	Male	*n*	Female
Inactive (*n* = 2084)	32.2 ± 1.6	836	32.2 ± 2.5	1248	32.3 ± 2.0
Somewhat active (*n* = 439)	402.1 ± 3.3	166	397.9 ± 5.2	273	404.6 ± 4.2
Active (*n* = 585)	731.0 ± 5.6	251	735.8 ± 8.8	334	727.5 ± 7.1
Very active (*n* = 612)	2333.2 ± 82.4	333	2499.1 ± 114.1	279	2135.2 ± 118.1

Values are expressed as means ± standard errors.

**Table 5 ijerph-17-05416-t005:** Odds ratio (95% CI) for MetS and MetS components according to physical activity levels.

Physical Activity Group	MetS	High Waist Circumference	High Triglycerides	Low HDL-C	High Blood Pressure	High Glucose
Inactive (*n* = 2084)	1.00 (reference)	1.00 (reference)	1.00 (reference)	1.00 (reference)	1.00 (reference)	1.00 (reference)
Somewhat active (*n* = 439)	1.03 (0.83–1.29)	0.91 (0.74–1.14)	0.96 (0.77–1.21)	0.88 (0.72–1.08)	1.08 (0.87–1.33)	0.99 (0.80–1.23)
Active (*n* = 585)	0.81 (0.66–0.98) *	0.80 (0.66–0.96) *	0.96 (0.79–1.18)	0.86 (0.72–1.04)	0.88 (0.73–1.06)	0.86 (0.72–1.04)
Very active (*n* = 612)	0.72 (0.59–0.88) **	0.77 (0.64–0.93) **	0.79 (0.65–0.97) *	0.64 (0.53–0.77) ***	0.82 (0.69–0.99) *	0.85 (0.71–1.02)

Data presented as odds ratio (95% confidence intervals (CIs)). All ORs are adjusted for age, sex, smoking state, and alcohol. OR = odds ratio, MetS = metabolic syndrome, HDL-C = high-density lipoprotein cholesterol. * *p* < 0.05 vs. reference, ** *p* < 0.01 vs. reference, *** *p* < 0.001 vs. reference.

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
