# Peer review of "Sex-Specific Energy Intakes and Physical Activity Levels According to the Presence of Metabolic Syndrome in Korean Elderly People: Korean National Health and Nutrition Examination Survey 2016–2018"

_ijerph, 2020, doi:10.3390/ijerph17155416_

Round 1
Reviewer 1 Report
This study aimed to analyze the differences in energy intake and physical activity (PA) levels according to sex and the presence of metabolic syndrome (MetS) among elderly people in Korea. With a very large group of elderly people. Therefore, this study is likely to be largely unique.
The paper is well structured and written without any digressions. The statistics and findings are presented in a clear and concise way, which makes it easy to understand.
However, there are two comments to the authors:
1. in order to better understand the ANOVA statements, Eta squared and the power should be calculated.
2. the discussion should focus on the transferability of the results, which are mainly influenced by people of Korean descent, to other population groups
Author Response
Response to Reviewer Comments
This study aimed to analyze the differences in energy intake and physical activity (PA) levels according to sex and the presence of metabolic syndrome (MetS) among elderly people in Korea. With a very large group of elderly people. Therefore, this study is likely to be largely unique.
The paper is well structured and written without any digressions. The statistics and findings are presented in a clear and concise way, which makes it easy to understand.
However, there are two comments to the authors:
1. in order to better understand the ANOVA statements, Eta squared and the power should be calculated.
-> Thank you very much for your comments. Eta squared and statistical power was calculated and added to the table as proposed by the reviewer.
2. the discussion should focus on the transferability of the results, which are mainly influenced by people of Korean descent, to other population groups
-> Thank you very much for your comments. I added the characteristics of not only domestic but also other populations to the discussion.

Reviewer 2 Report
The introduction, methods and results are well written. Some of the papers quoted in the discussion would be better placed in the introduction. An opportunity is missed to report the quality of food and micronutrient as well as macronutrient intake. This is a cross-sectional study that can only show an association. The discussion needs reworking accordingly with a careful discussion of limitations. It would be much improved by a discussion of prospective or interventional literature to support the findings.
In the methods it states that 20,659 individuals aged >65 years completed the survey and examination. However, from the flow diagram it seems that 15,703 were removed, as they were <65 years. Is the statement in the text an error?
In that flow diagram, it would be helpful to break down those excluded because of cancer separate to those with missing data.
The data on physical activity and dietary intake were derived from retrospective recall and so requires careful clarification of the anticipated limitations in the discussion. Please reference the validated method used for estimation of METS based on activity and classification by group in section 2.4.
In table 1 (and other tables) all data are reported to 2 decimal places, but this is unlikely to represent the accuracy of the parameters, for example, blood pressure cannot be measured to the nearest 0.01mHg. Please adjust all parameters to clinically meaningful measures of accuracy.
Would the demographics such as BMI 24kg/m2 be representative of the Korean general population, or was it a non-representative group that entered the study?
Was the data normally distributed? Please state the covariates adjusted for in the logistic regression. What were the conclusions from this analysis?
It would be usual to refer to ‘blood glucose’ rather than ‘blood sugar’.
Were people with diabetes mellitus included in the data set? Could those people with MetS and diabetes mellitus be limited in their physical activity by complications of their disease? Would the subset with diabetes be instructed to follow a different diet making it a cause of the change in energy intake rather than an effect? Do the subset of MetS patient with FG>100mg/dl behave differently from those without dysglycaemia, as some (but not all) of these)may have diabetes?
Can it be said that the elderly women had a ‘nutritional imbalance’ based on a high amount of carbohydrate and a low amount of fat. Some assessment of the quality of the carbohydrate would be needed, and a low saturated fat diet may be appropriate? Did micronutrient intake meet standards? Was vitamin D measured as a potential confounder?
Should overweight people with MetS increase their food intake as you suggest, is there evidence for this? It is mentioned in the discussion that the increased incidence of MetS is caused by Westernized eating habits. Can this be assessed from the available data/ Were those with MetS more likely to eat a Western diet?
It cannot be said from this data that ‘to prevent MetS and MetS related variables, PA above the active or very active level are required’ or that these results suggest that ‘to prevent MetS in elderly people PA levels should be increased and balanced nutrition should be assured’. Whilst this may be true, this is a retrospective study that can only show an association and not causality.
Therefore a review of available literature on randomized or prospective lifestyle interventions in the elderly would be helpful and support any conclusions made from the cross-sectional data.
Author Response
Response to Reviewer Comments
The introduction, methods and results are well written. Some of the papers quoted in the discussion would be better placed in the introduction. An opportunity is missed to report the quality of food and micronutrient as well as macronutrient intake. This is a cross-sectional study that can only show an association. The discussion needs reworking accordingly with a careful discussion of limitations. It would be much improved by a discussion of prospective or interventional literature to support the findings.
In the methods it states that 20,659 individuals aged >65 years completed the survey and examination. However, from the flow diagram it seems that 15,703 were removed, as they were <65 years. Is the statement in the text an error?
-> As you pointed out, it's an error in the sentence. Of the total 20,659 people, 15,703 people under the age of 65 were excluded. I modified it based on this.
In that flow diagram, it would be helpful to break down those excluded because of cancer separate to those with missing data.
-> As you pointed out, we each recorded cancer and the number of data missing persons other data, on the diagram flow.
The data on physical activity and dietary intake were derived from retrospective recall and so requires careful clarification of the anticipated limitations in the discussion. Please reference the validated method used for estimation of METS based on activity and classification by group in section 2.4.
-> As you pointed out, we added a reference.
In table 1 (and other tables) all data are reported to 2 decimal places, but this is unlikely to represent the accuracy of the parameters, for example, blood pressure cannot be measured to the nearest 0.01mHg. Please adjust all parameters to clinically meaningful measures of accuracy.
-> Thank you very much for your comments. We corrected it to clinically meaningful measures of accuracy.
Would the demographics such as BMI 24kg/m2 be representative of the Korean general population, or was it a non-representative group that entered the study?
-> Thank you very much for your comments. The 2016-2018 National Health and Nutrition Survey found that 92% of the total number of older aged 65 or older was in the range of 18.5 to 30 kg/m2, and the previous studies Kim & Kim (2018) and Hong (2019) also showed that the average BMI of elderly was 24 kg/m2 levels. Therefore, the BMI mean of the people in this study is estimated to correspond to the general population characteristics.
- Kim, S., & Kim, D. I. (2018). Association of regular walking and body mass index on metabolic syndrome among an elderly Korean population. Experimental gerontology, 106, 178-182.
- Hong, S. (2019). Association of relative handgrip strength and metabolic syndrome in Korean older adults: Korea National Health and Nutrition Examination Survey VII-1. Journal of Obesity & Metabolic Syndrome, 28 (1), 53.
Was the data normally distributed? Please state the covariates adjusted for in the logistic regression. What were the conclusions from this analysis?
-> In this study, the distribution of normality was confirmed and described in the study method. The covariate were analyzed by correcting age, gender, drinking, and smoking conditions, and are presented in the bottom 5 of Table. The conclusion of this study suggests that the incidence of metabolic syndrome is reduced when physical activity is performed over 500 MET per week in cross-section, and the risk of metabolic-related variables can be reduced when it is above 1000 MET.
It would be usual to refer to ‘blood glucose’ rather than ‘blood sugar’.
-> I revised it as you pointed out.
Were people with diabetes mellitus included in the data set? Could those people with MetS and diabetes mellitus be limited in their physical activity by complications of their disease? Would the subset with diabetes be instructed to follow a different diet making it a cause of the change in energy intake rather than an effect? Do the subset of MetS patient with FG>100mg/dl behave differently from those without dysglycaemia, as some (but not all) of these)may have diabetes?
-> Thank you very much for your comments. The subjects of this study include elderly people with diabetes. However, there were no elderly people with metabolic syndrome or diabetes who had restrictions on physical activity due to complications. However, in the prior study, the subjects with diabetes had low physical activity, and in the prior study, it is reported that those with diabetes have high carbohydrates and low intake of fat and protein.
In this study, we failed to identify the relationship between nutrition and physical activity according to diet control or dyslipidemia among those with metabolic syndrome pointed out by the reviewer. However, prior studies have shown that more than half of those diagnosed with diabetes consume more than 70 percent of calories from carbohydrates and avoid foods high in fat.
Can it be said that the elderly women had a ‘nutritional imbalance’ based on a high amount of carbohydrate and a low amount of fat. Some assessment of the quality of the carbohydrate would be needed, and a low saturated fat diet may be appropriate? Did micronutrient intake meet standards? Was vitamin D measured as a potential confounder?
-> Thank you very much for your comments. As you pointed out, it is not possible to say that there was an imbalance in nutrition due to carbohydrates and fat, but in a prior study, an elderly person with metabolic syndrome reported a low intake rate in all nutrients other than carbohydrate protein fat. Given the low intake of carbohydrates, fats, and proteins in older people with metabolic syndrome shown in this study, it is believed to be related to a lack of nutrition similar to prior studies. Vitamin D was not measured in the national health and nutrition survey used in this study, so we couldn't check it. Please reconsider.
Should overweight people with MetS increase their food intake as you suggest, is there evidence for this? It is mentioned in the discussion that the increased incidence of MetS is caused by Westernized eating habits. Can this be assessed from the available data/ Were those with MetS more likely to eat a Western diet?
-> Thank you very much for your comments. In the discussion section of this study, we removed the content that increases intake and that the incidence of metabolic syndrome is due to Western eating habits.
Unlike Westerners, this study revised the content that it is important to properly consume various nutrients as a way to solve problems caused by malnutrition, not by excess nutrition.
It cannot be said from this data that ‘to prevent MetS and MetS related variables, PA above the active or very active level are required’ or that these results suggest that ‘to prevent MetS in elderly people PA levels should be increased and balanced nutrition should be assured’. Whilst this may be true, this is a retrospective study that can only show an association and not causality.
Therefore a review of available literature on randomized or prospective lifestyle interventions in the elderly would be helpful and support any conclusions made from the cross-sectional data.
-> Thank you very much for your comments. I deeply empathize with what the reviewer pointed out. It is not enough to suggest that elderly people with metabolic syndrome should increase their physical activity cross-sectionally and take balanced nutrition. It is deemed necessary to study various causal relationships such as the lifestyle and social conditions of the elderly.

Round 2
Reviewer 2 Report
Thank you for addressing the comments. I have minor suggestions:
Line 99: The added sentence ‘Among them, 15,703 people <65 years were excluded, while 4,956 were aged’ does not make sense, please review.
There are language, grammar and syntax errors in the manuscript editions shown in red font. This does impact on the ability of the reader to understand the discussion.
Line 265: Please review the two new sentences for readability as this could be stated more clearly and succinctly.
Line 290: Please review the new paragraph for readability, you may have included typographical errors e.g. ‘’variety of corrosion’; ‘domestic elderly people’? The language used needs review to ensure it is easy to read.
Line 310: Please review the new paragraph for grammar e.g. ‘basic data onto preparing countermeasures’; ‘combining the results of this study and prior stud, the nutritional status of the elderly with MetS becomes even worse’ etc.
Author Response
Response to Reviewer Comments
Thank you for addressing the comments. I have minor suggestions:
Line 99: The added sentence ‘Among them, 15,703 people <65 years were excluded, while 4,956 were aged’ does not make sense, please review.
-> Thank you very much for your comments. I revised and supplemented according to the reviewer comments. The revised contents are as follows: Among them, 15,703 people under 65 years of age were excluded, and over 65 years of age were 4,956 people.
There are language, grammar and syntax errors in the manuscript editions shown in red font. This does impact on the ability of the reader to understand the discussion.
-> Thank you very much for your comments. I revised and supplemented the overall contents of the language, grammar and syntax errors according to the reviewer comments.
Line 265: Please review the two new sentences for readability as this could be stated more clearly and succinctly.
-> Thank you very much for your comments. I revised it clearly and succinctly according to the reviewer comments.
Line 290: Please review the new paragraph for readability, you may have included typographical errors e.g. ‘’variety of corrosion’; ‘domestic elderly people’? The language used needs review to ensure it is easy to read.
-> Thank you very much for your comments. I corrected the typographical errors according to the reviewer's comments and corrected them to make them easier to read.
Line 310: Please review the new paragraph for grammar e.g. ‘basic data onto preparing countermeasures’; ‘combining the results of this study and prior stud, the nutritional status of the elderly with MetS becomes even worse’ etc.
-> Thanks for your comments. I modified it according to the reviewer comments.